# Vine and Wine Sustainability in a Cooperative Ecosystem—A Review

Agostinha Marques [1,2] and Carlos A. Teixeira [2,*]

1 Adega Cooperativa de Favaios, 5070-265 Vila Real, Portugal; agostinhamarques@adegadefavaios.com.pt
2 Centre for the Research and Technology of Agro-Environmental and Biological Sciences (CITAB), Institute for Innovation, Capacity Building and Sustainability of Agri-Food Production (Inov4Agro), University of Trás-os-Montes and Alto Douro (UTAD), 5000-801 Vila Real, Portugal
* Correspondence: cafonso@utad.pt

**Abstract:** The world is changing, and climate change has become a serious issue. Organizations, governments, companies, and consumers are becoming more conscious of this impact and are combining their forces to minimize it. Cooperatives have a business model that differs from those in the private or public sector. They operate according to their own principles of cooperation, which makes it difficult to obtain results that are in harmony with the objectives of the organization and the cooperative members. However, they are also aware of climate change because their businesses are directly affected. Thus, in this review, we have tried to answer the following questions: What is necessary to meet the sustainability goals? Are wine cooperatives competitive in the context of the global market? How can we respond to the challenges of environmental sustainability while maintaining wine quality standards and economic profitability? What are the economic and social impacts of reducing the carbon footprint of cooperatives and their members?

**Keywords:** wine cooperative; sustainability; benchmarks

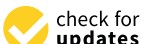



## 1. Introduction

In 2015, the United Nations established an agenda for 2030 to 2050 and defined the 17 Goals for Sustainable Development, which aim to improve living conditions; combat poverty, hunger, and social inequity; promote access to water, health, and education; combat climate change; and protect the environment [1–3]. In this respect, governments, companies, and organizations have been looking for ways to respond to the United Nations' challenge. Europe, for its part, has taken the leading role in combating climate change, particularly in the food sector, with the creation of the European Green Deal, whose goals include agriculture that is environmentally sustainable and a fair and healthy food system [4].

Wine production is one of the oldest economic activities, and environmental factors have always affected grape production, forcing people to select grape varieties according to the terroir and the soil in order for greater efficiency [5]. The cultivation of wine has transformed landscapes and has become one of the sectors that contributes most to the economic and social sustainability of communities. It is an integral part of culture, providing many experiences, encouraging tourism, and being a source of pride for communities [2,6].

One of the sectors that most contributes to greenhouse gas emissions is agriculture, with the wine sector accounting for 0.3% of global GHG emissions (considering a bottle of wine leaving a cellar), and promoting sustainable environmental behavior has consequently been the subject of certain policies [7]. Viticulture has a large impact on the environment, as the use of chemical products, soil tillage, irrigation, soil management, and mechanization are all responsible for GHG emissions [4,8].

In 2004, the OIV defined viticultural sustainability as a global strategy for grape and wine production which contributes to the economic sustainability of communities by producing quality products and practicing responsible viticulture. Sustainable viticulture is concerned with risks to the environment, product safety, and consumer health, as well as

valuing local heritage, history, landscape, and culture [9–11]. There is a growing commitment in agriculture to more sustainable practices [12], not only because of the economy, but also for environmental reasons and the legacy for future generations. Sustainability and the efficient use of environmental, social, and economic resources are becoming increasingly important to wine consumers and winemakers. This is clear from the way that markets and consumers prefer products produced and labeled according to "sustainability indicators or terms", such as organic, sustainable, natural, free, ecological, etc. [9], because for the consumer, the term "sustainable" is associated with the environment and their carbon footprint. Governments, for their part, have been trying to impose measures that encourage consumers to choose products that are more sustainable, for example, by applying environmental taxes (on carbon) [7] or, in the case of monopolies, restricting products that do not meet sustainability standards.

Wine cooperatives are considered to be organizations with sustainable social and economic development as some of their multiple roles and objectives [13], and they feel pressure not only from consumers, but also from governments and monopoly markets. Ziegler [14] argues that wine cooperatives should have objectives and strategies to ensure circular social and ecological sustainability.

Growing pressure for political reasons and customers looking for sustainable products [5,15] have created the need for winegrowing organizations to develop indicators aimed at the efficient use of water, production methods, the use of phytopharmaceuticals, energy efficiency in the vineyard and winery, the promotion of clean energy rather than fossil fuels, waste management, community impact, and employee well-being [5,16]. This is because, for them, and in contrast to the consumer, sustainability is not only environmental, but also economic and social [9]. In addition, consumers are becoming increasingly aware of the need to be more sustainable, and wine producers need to implement sustainable practices in order to stand out in a market with growing competition [2,6,17]. This has led to the creation of various sustainable certification programs in winegrowing, which winegrowers have tried to adopt. Cooperatives, formed by small winegrowers, most of them with very limited literacy, cannot impose these rules; cooperatives will have to create tools to encourage their members to adopt sustainable practices in response to market demands.

In our review of the literature, we found that guidelines have been defined by the authors mentioned in Table 1. This was the first observation that research has only focused on one aspect of sustainability, and in the case of cooperatives, the focus is on economic and social sustainability.

**Table 1.** Literature review.

| Ref. | Authors | Year | Country | Relevant Information |
|---|---|---|---|---|
| [1] | Chabin et al. | 2023 | France | Sustainability; 17 Sustainable Development Goals; Economy; Environment; Resources |
| [2] | Ferrer et al. | 2022 | Spain | Economy; Environment; Resources |
| [4] | Nazzaro et al. | 2022 | Italy | Innovation; Sustainability; Cooperatives; Governance; European Green Deal |
| [5] | Tsalidis et al. | 2022 | Greece | Organic; Viticulture; GHG emissions; Carbon Footprint |
| [6] | Martínez-Falcó et al. | 2023 | Spain | Sustainable Development Goals; Wine Industry |
| [7] | Soregaroli et al. | 2021 | Italy | Carbon Footprint; Climate Change; Wine Consumers |
| [9] | Lamastra et al. | 2016 | Italy | Vineyard Sustainability; Indicators; Environmental |
| [10] | Marras et al. | 2015 | Italy | Vineyard Management; Carbon Footprint; Agriculture; GHG Emissions |
| [11] | Casolani et al. | 2022 | Italy | LCA; Wine Sustainability; Environment Sustainability |

## 2. Cooperative Ecosystem

In 1852, Great Britain declared the cooperative a business for the first time [18]. This shows the cooperative tradition in Europe [19]. According to the "Declaration of Cooperative Identity" defined by the International Cooperative Alliance in 1995, "a cooperative is

an autonomous association of persons voluntarily united to meet their common economic, social and cultural needs and aspirations through a jointly owned and democratically controlled enterprise" [14,20,21]. Article 2°, paragraph 1 of the Portuguese Cooperative Code defines a cooperative as "collective and autonomous persons, free constituted, with variable capital and composition, which, through the cooperation and mutual help of their members, in compliance with the cooperative principles, aim, on a non-profit basis, to satisfy their economic, social or cultural needs and aspirations" [20,21]. Cooperatives are governed by seven main principles: voluntary and free membership, democratic management by members, economic participation by members, autonomy and independence, education, training and information, cooperation between cooperatives, and interest in the community [21,22]. In other words, cooperatives are socially based people's enterprises [22], and stand out for promoting social equality, community development, and the well-being of their members [20]. We can conclude that cooperatives are the best business model for local development, considering the cooperation between citizens and local, regional, and national organizations [20].

Lately, there has been growing interest in the cooperative model, as this business model has proved to be more resilient in times of prolonged economic crises than capitalist companies [13]. Cooperatives favor the maintenance of jobs, preferring to reduce salaries, and the distribution of surpluses is more balanced to meet needs in times of crisis [13]. Historically, there has been an increase in the creation of cooperatives in times of economic and social crisis [20], such as in the production of the liqueur muscatel in Portugal in the 1950s. Ziegler [14] has conducted a study showing that cooperatives are fundamental to the circular economy and its incorporation into regional economies, concerning revalorization, production, consumption, and lasting use.

These organizations are more sensitive to environmental, social, and economic issues due to their cooperative values [12]. Since equality, community development, the well-being of their members, and combating exclusion and poverty among the most disadvantaged classes are at the genesis of the creation of cooperatives, they are an alternative business model to capitalism [13]. This business model helps small producers to create scale, i.e., they are able to sell their products more easily as they gain the capacity to negotiate by volume [20]. However, there are also weaknesses, since a cooperative demands the acceptance of all the production of its cooperative members, without taking into consideration quality or production methods, and can only impose a few rules that benefit those who comply to the detriment of those who do not [19].

Figueiredo [20] defined cooperatives and cooperative members as "social entrepreneurs" who are orientated towards financial independence and sustainable entrepreneurship to create social value for the less privileged. We can therefore say that cooperatives enable the creation of stronger and more sustainable local economies because they reinvest profits, without forgetting social values and their mission [20].

As cooperatives are solutions for local development, agricultural cooperativism is very much in the spotlight, especially when we look at production. According to Figueiredo [13], 41% of the wine produced in Portugal is made by cooperatives, and the numbers are even more impressive when it comes to milk, which accounts for around 62%. This is why agricultural cooperatives are so important, given that they operate at a rural level and contribute to the conservation of these environments and the environment in general [20]. However, like other companies, they must be competitive and create value in order to become economically, socially, and environmentally sustainable [20].

Climate change has been challenging companies to take urgent action to maintain their competitive edge [19]. Some studies show that cooperatives are more proactive on environmental issues than private companies [12], but there is no evidence of their application in agricultural practices, such as in reducing their carbon footprint, water footprint, use of fossil fuels, etc., since there is a lack of documentation or sustainability reports by cooperatives; these reports could not only show their commitment and sustainability strategy, but could also be seen as an internal learning mechanism [14].

Figueiredo is one of the most widely published authors in the field of cooperatives and their dynamics. Analyzing the articles by Ritcher and Figueiredo has provided a better understanding of the fundamentals and the cooperative business model (Table 2).

**Table 2.** Cooperativism literature review.

| Ref. | Authors | Year | Country | Relevant Information |
|---|---|---|---|---|
| [12] | Calle et al. | 2020 | Spain | Cooperatives; Environmental; Wine Sector |
| [13,20] | Figueiredo et al. | 2018 | Portugal | Cooperative; Sustainability; Social; Economic; Society Development |
| [14] | Ziegler et al. | 2023 | Canada | Cooperatives; Circular Economy; Business Model; Social Economy |
| [18,19] | Ritcher et al. | 2021 | Germany | Sustainable Management; Cooperatives; Cooperative Values; Social Capital |
| [21] | Ramos et al. | 2023 | Portugal | Cooperatives; Democracy; Governance |

## 3. Difficulties in Respect to Responses from Cooperative Members

The cooperative model depends on the ability of cooperatives to satisfy the ambitions of their members, which sometimes do not meet the principles of cooperativism due to the external and internal pressures that management can face [20]. This disruption can lead to a loss of cooperative identity [20]. For this reason, when results are equal to or better than expected, satisfaction is high and fundamental to maintaining trust, cooperation, and commitment between everyone, cooperatives and cooperators, reducing disputes [20]. In addition, through the difficulties inherent in cooperativism, the wine sector suffers from the effects of demographics and land abandonment. According to Figueiredo's research [13], the average age of cooperative members is around 60, they are mostly men, and they have low literacy levels. They are also resistant to change, and issues of efficiency and performance are of lesser importance. The great challenge for cooperatives lies in their ability to attract younger members to maintain the sustainability of the organization [13].

Another difficulty is related to one of the cooperative principles, freedom, i.e., there is an "open door" policy, which enables the free entry but also the free exit of members, which leads to problems of opportunism and lack of commitment [20]. Differences between members, like quantity, grape production as a main or secondary activity, acceptance of risk, and organization, contribute to a high degree of heterogeneity between members, which slows down decision-making [18]. Due to this heterogeneity, the challenge is to persuade members to apply sustainability measures [19].

However, it is not only the cooperative members who create difficulties. One of the biggest problems is caused by the cooperative itself: the payment periods for cooperative members are long, never less than 90 days, and often more than two years, which is one of the main reasons why cooperative members leave, as they need immediate liquidity [13].

An advantage of the cooperative system is that when the governance model is oriented to innovation and development, this allows access to innovative technologies and techniques, such as precision agriculture [4]. As well as promoting knowledge, this can make investments in technology accessible to cooperative members, since individual investment would be economically unviable. However, this can be criticized due to differences in objectives between management and cooperative members; one of the most common situations is production vs. quality, with the cooperative looking for quality and the cooperative members seeking production [4].

It is difficult for farmers to measure all the indicators they need to take advantage of in a sustainability framework [23]. The lack of a clear standardization of indicators leaves winegrowers in doubt about which indicators are essential for understanding their company's level of sustainability, and in responding to market demands [24,25] and determining how to do so. The process is more complicated when applied to wine cooperatives. In a private company, the management board easily defines the objectives to be met by the organization, while in the case of cooperatives, the decision-making capacity of the

management board is more limited not only because it is an elected position, but also because of the time limitation for implementing long-term objectives [19]. This difficulty is compounded by the fact that, in general, investments in sustainable measures have a long-term effect and the winegrower needs funding in the short-term, so money is more important [18]. Communication between the board and the members is essential; it is important that the members understand that consumers are now willing to pay more for sustainably produced wine [18].

Faced with the current situation and the analyses carried out in this study, it is necessary to provide cooperatives with tools that support them in materializing their values and responding to the markets [12,18], and that allow the cooperative to prevail in the long-term.

Understanding the dynamics of cooperatives requires an understanding of their strengths and weaknesses. Since cooperatives are created to help a large and heterogeneous number of individuals, this creates many challenges that are not found in private companies. In Table 3, the authors of this study have gathered some information, but many questions remain unanswered.

**Table 3.** Difficulties with cooperativism, literature review.

| Ref. | Authors | Year | Country | Relevant Information |
|------|---------|------|---------|----------------------|
| [23] | Withehead | 2017 | New Zealand | Sustainable Development; Sustainability Assessment; Wine |
| [24] | Borosato et al. | 2020 | Italy | Viticulture; Sustainability; Innovation |
| [25] | Merli et al. | 2018 | Italy | Indicators; Environmental Management Systems; Sustainability |

## 4. Different Sustainability Benchmarks

Over the years, several sustainable certification benchmarks have emerged which differ from organic, biodynamic, and biological certification [15]. Although they have the same objectives, they are different in terms of methodology [15]. This diversity of benchmarks for certification in the wine sector [25] has led to some markets (export, national) feeling the need to create a set of rules in which the sustainability indicators fit in with greater or lesser importance, as is the case with SystemBolarget, created in Sweden [26], and Sonae's Producers Club in Portugal.

In the wine sector, there are various models of certification. These can involve the certification of vineyards, wineries, or both [15]. For example, although organic farming has a positive impact on the environment, it has little focus on sustainability [15]. ISO 14001 was designed in the 1990s and is an environmental management system with an auditing program. It is voluntary and includes all economic areas, including agriculture and more specifically the wine sector [15].

In order to regulate the sector in 2020, the OIV (International Organization of Vine and Wine) worked on a guide for implementing the principles of sustainability in viticulture [27]. The sustainable certifications that have subsequently emerged use the OIV's guidelines in this document as a basis [28]. However, while the key indicators are common across the different benchmarks for certification, the ways in which they are described vary; for example, in calculating the carbon footprint, energy consumption, impact of the carbon footprint on soil, GHG [28], water footprint [29], etc. However, the indicators usually tend to be more descriptive than analytical, making it difficult to determine the questions to be measured and their answers, which is a weakness of the system [23].

Sustainable certification benchmarks in the wine sector first started in New Zealand in 1997 with the "Sustainable Winegrowing New Zealand" program [30]; others have been emerging [15], most recently in Portugal with ViniPortugal's "National Reference for Sustainability Certification in the Wine Sector" in 2022 [31] and the IVDP's "Sustainability Manual for the Douro Wine Region" in 2023 [32] (Figures 1 and 2). Portugal currently

has the Alentejo (PSVA), launched in 2015 and promoted by the Alentejo Regional Wine Commission [33].

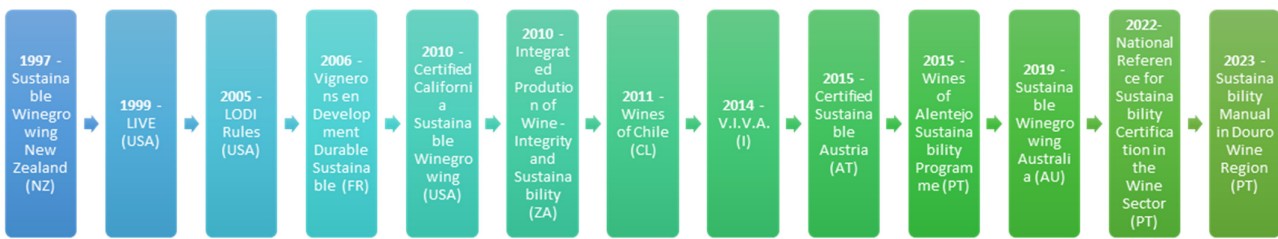

**Figure 1.** Timeline for the creation of the different sustainable certification models for the wine sector.

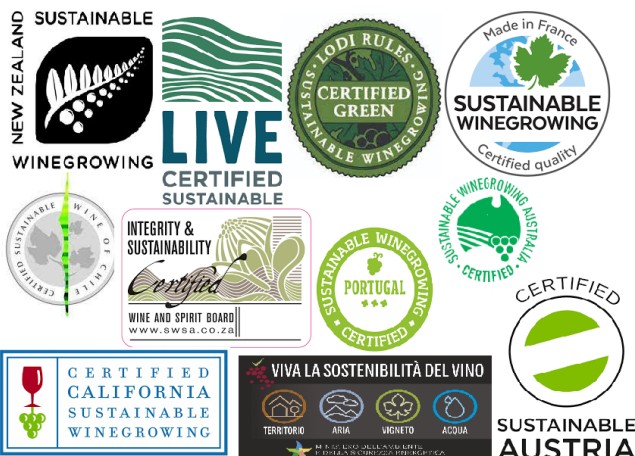

**Figure 2.** Labels associated with different sustainable certification models in the wine sector.

Some of the best-known sustainable certification benchmarks for the wine sector, created specifically for the vine and wine sector, are described below (Figure 1).

### 4.1. Sustainable Winegrowing New Zealand (1997)

In 1995, the New Zealand Winegrowers Association started the "Sustainable Winegrowing New Zealand" program, its success being such that in 1997 they began the process of certifying producers at a national level [15]. It is a national program with a sustainability label, financed by a tax on the sale of grapes and wine and by the cost of certification. The process has 62 chapters, based on various indicators such as biodiversity, soil, air, water, energy, chemicals, by-products, people, and the economy. To obtain certification, it is necessary to have an audit carried out by an independent auditor [15,30]. One of the main criteria for the label is that the grape and wine are produced 100% sustainably [30]. New Zealand's progress has given it a competitive advantage over other winegrowing regions in the world [15].

### 4.2. LIVE (1999)

LIVE is the first North American certification benchmark to originate in Oregon. A nonprofit organization, the LIVE program was created in 1999, based on the indicators of the International Organization of Biological Control (IOBC) for Integrated Pest Management (IPM). Today, the program is not so focused on chemical products; environmental, economic, and social indicators have been added [15,34].

### 4.3. LODI Rules (2005)

The LODI Rules certification came into being in 2005, but its basis was created in 1992 when the Lodi Winegrape commission and the University of California State Agricultural Extension created a document based on sustainable practices. For a producer to use the



LODI Rules label, 85% of the grapes used must be certified and the minimum score for each chapter is 50%, with a minimum total score of 70% for certification. The indicators are divided into six chapters: economy, human resources, ecosystems, soil, water, and pests [15,35].

### 4.4. Sustainable Development for Wine Growers (2006)

This is the first certification benchmark in Europe, which originated in France and was launched in 2006. It is based on four pillars: environmental preservation, wine quality, society factors, and a fair price for the consumer. To obtain certification, the producer must be a member of an association, fulfil 37 indicators, and obtain at least 50% [15,36].

### 4.5. Certified California Sustainable Winegrowing (2010)

In 2001, California developed a Sustainable Winegrowing program and in 2003 it was included in the California Sustainable Winegrowing Alliance (CSWA). In 2010, Certified California Sustainable Winegrowing was born with the aim of training and providing growers with tools to improve sustainable winegrowing practices. Today, it focusses on more transversal sustainability, with the main pillars focusing on the environment, economy, and society factors. To obtain certification in viticulture, 50 indicators must be met, and wine production must meet 32 indicators. Like other programs, after a self-assessment, an audit is carried out by an independent auditor [15,37].

### 4.6. Integrated Production of Wine—Integrity and Sustainability (2010)

This was perhaps the first sustainability program, as in 1998 the Integrated Production of Wine (IPW), run by the South African government, was established. However, it was only in 2010 that the sustainability label and certificate were created, which is why the start is attributed to 2010. This is one of the few programs that has no cost to the producer and therefore also has one of the highest uptakes. To obtain certification, producers must have a minimum of 162 points out of 270, in 27 indicators. In the case of wineries, 93 points out of 155 points are required in 31 indicators [15,38].

### 4.7. Wines of Chile (2011)

Chile's wine sustainability certification program was created in 2011 with the impetus of the wine industry and is managed by a non-profit organization. It focuses on the three pillars of sustainability, not just the environmental pillar, applied to the vineyard, winery, bottling, and human resources. It mainly seeks to reduce the risks of the production system and the vulnerability of the sector to environment and climate change [15,39].

### 4.8. V.I.V.A (2014)

The V.I.V.A. program appeared as a pilot project of the Italian government, Ministry of Environment, Land and Sea in 2011 and the first certification was made in 2014. Certification is financed by the government and aims at the sustainability of the sector and adding value to the certified product. It focuses on four chapters: water, vineyard, air, territory. It is also the first program to make publicly available the results of the audits made by an independent auditor, making it a transparent program [15,40].

### 4.9. Certified Sustainable Austria (2015)

Austria has taken existing programs and adapted them to its reality. Its program was created in 2015 by the Austrian Winegrowers Association and is national in scope. Austria is one of the European countries with the largest area of vineyard certified as organic, so its adaptation was easy and in the first year 23 wineries were certified. Certification works on a traffic-light scheme, with green being the most sustainable. Producers respond in an online tool that can be consulted by the consumer in a model of transparency, like the Italian program [15,41].

### 4.10. Wines of Alentejo Sustainability Program (2015)

This was the first sustainability program created in Portugal and adapted to the Alentejo region. The program was initiated in 2013 by the organization that controls the wines of Alentejo (Comissão Vitivinícola do Alentejo), and it was inspired by the California model, CSW, due to the similarities in production, climate, and terroir. It is divided into three sectors: vineyard, winery, and vineyard and winery. It has 18 chapters and 171 indicators, based on four global pillars. The first is supervision, management, and quality; the second is social, the third is environmental; and the fourth is exclusive requirements. For wine certification, 60% of the vineyard area must be registered in the PSVA. It has a scale of levels that starts at initial, where growers must achieve 60%, followed by intermediate, with the last being developed [33].

### 4.11. Fresh Australian Wine Industry Standard of Sustainable Practice (2020)

Launched in 2019, this program is based on the "Sustainable Australia Winegrowing" (2011) and Entwine Australia programs and was revised in 2020. In 2020, the benchmark was categorized into two parts, viticulture and winery, and was renamed. It is a national program aimed at winegrowers and winemakers. The main pillars are social, economic, and environmental, with landscape and soil, water, people, the economy, biodiversity, energy, and waste being the most prominent [15,42].

### 4.12. National Reference for Sustainability Certification in the Wine Sector (2022)

This program, launched in 2022, is one of the most recent sustainability programs in Portugal. It was developed by two public organizations, one for control and the other for promotion, i.e., Instituto da Vinha e do Vinho (IVV) and ViniPortugal, respectively, based on programs already implemented in other regions of the world, such as the Alentejo program (PSVA), California Sustainable Winegrowing (CSW), LODI Rules, Bodegas Argentinas, Sustainable Winegrowing Australia, etc. It is based on four pillars, which are management and continuous improvement, environmental, social, and economic, which are divided into 86 indicators spread over 17 chapters. To obtain certification, 50% of the grapes must meet the minimum requirements of the program. The classification corresponds to letters, the lowest being C (ranging from 50% to 65%) and the highest being A (more than 85%) [31].

### 4.13. Sustainability Manual for the Douro Wine Region (2022)

The more recent sustainability program in Portugal is the Sustainability Manual for the Douro Wine Region, developed by IVDP and the Faculdade de Ciências da Universidade do Porto, which is currently under public consultation. It is a program very similar to the Californian CSW, which works on a colored traffic-light system, like Austria's program. The scoring criteria take into account the size of the companies in terms of area, volume of liters, turnover, and number of employees. However, it has one of the lowest acceptance levels; from 33%, it already has a D classification, the remaining levels being similar to the ViniPortugal program. This program focuses only on one region, which is the Douro. Like the other programs, it addresses the SDGs and is based on the main pillars: economy, social, environmental, and quality [28,32].

## 5. Different Sustainability Benchmarks

It is possible to analyze which indicators are the most important or eliminatory for each certification model. Furthermore, there has been an evolution in the certification models, with the most recent ones not only being more demanding, but also having more indicators aimed at economic and social sustainability (Figure 3). The first certification models focused more on vineyard, water, and soil aspects [15]. Biodiversity and water management are indicators mentioned in all of the sustainable certification models [10] (Figure 4). In New Zealand, Whitehead [23] analyzed the priority indicators for sustainability analysis and concluded that the water indicator is the most valued. This may be due to the notion that

it is a finite resource that is becoming increasingly scarce, with implications not only for agriculture, but also for everyone's day-to-day life.

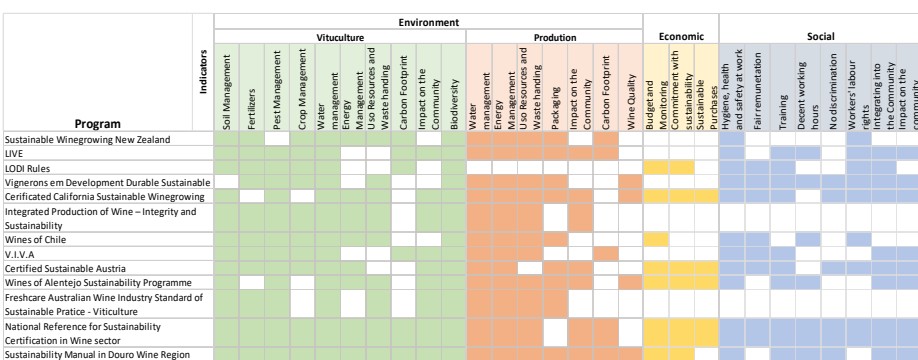

**Figure 3.** Sustainable certification benchmarks from the oldest to the most recent, showing the most important indicators for each benchmark (legend: green—indicators mentioned in the benchmarks for viticulture; orange—indicators mentioned in the benchmarks for wine production; yellow—economic sustainability indicators mentioned in the benchmarks; blue—social sustainability indicators mentioned in the benchmarks).

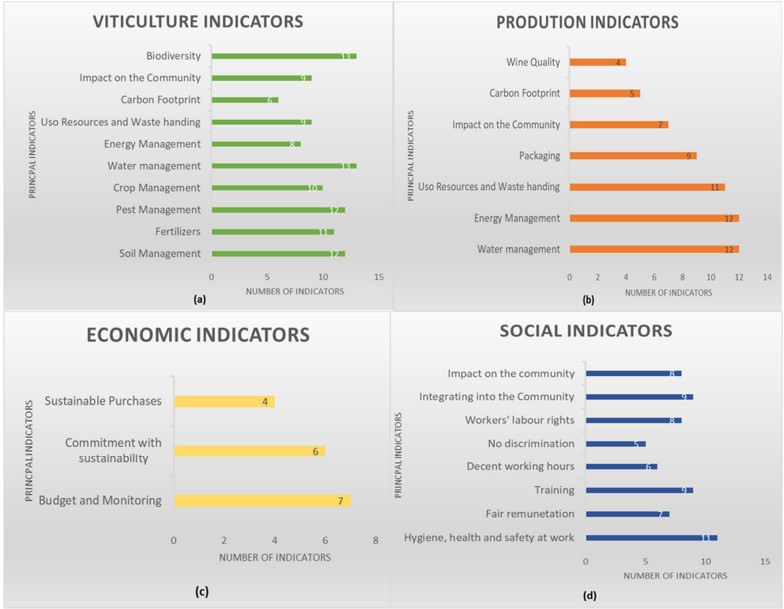

**Figure 4.** The most important sustainability indicators in the different sustainable certification benchmarks: (**a**) represents the environmental indicators for the vineyard (green color) and the number of benchmarks that measure them, and the most important are biodiversity and water management; (**b**) represents the environmental indicators for wine production (orange color) and the number of benchmarks that measure them, and again, water management is an important indicator, as is energy management; (**c**) represents the company's economic management indicators (yellow color) and the number of benchmarks that measure them, with budget and monitoring being the most relevant; and (**d**) represents social indicators (blue color) and the company's relationship with the community and the number of benchmarks that measure them, with hygiene, safety, and health at work being the most mentioned, but others also appear, such as training and integration into the community.

Biodiversity is approached in various ways. In older models, the focus was on maintaining the oldest and regional grape varieties, as well as the ecosystem. The most recent models focus on the vineyard's ecosystems, such as forest, riparian, small vegetation, and bird nesting sites, and the correct maintenance of these ecosystems. Mulch is becoming increasingly important [31–33,37]. In addition to increasing the soil's ability to retain water,

it is a shelter for pest predators and a source of nutrients for the plant, as well as reducing the invasion of undesirable weeds. A good mulch helps to reduce tillage and the use of insecticides, herbicides, and fertilizers, creating greater water retention in the soil and the prevention of soil leaching.

While all the models give importance to the social aspect, it can be seen that "Hygiene, Health and Safety at Work" is present in most of them, as is training. However, these indicators are legal requirements in Europe and the USA, so this is more a way of checking legal compliance, although it can also be seen as an opportunity for improvement.

Below are some graphs (Figure 4) showing which indicators are most relevant to the different benchmarks. Only the most relevant were selected and/or were an eliminating factor in certification.

In this set of graphs (Figure 4), the importance of environmental indicators is clear, especially in the vineyard. Economic indicators are only evaluated in a macro way, which encourages analyses in the direction of economic sustainability. Social indicators are becoming increasingly important, especially on the part of consumers. Consumers prefer products whose production respects human rights, such as fair wages, non-discrimination, and social equity [9,26]. Interaction with the local community is also valued, in terms of the circular economy and minimizing the environmental impact of the activity [5,6,16].

The carbon footprint is an indicator that is not directly addressed in some of the certification benchmarks, but most organizations have online availability so that producers can calculate it [32,34,35,37,40]. However, this is the indicator that consumers recognize most easily, perhaps because it is applicable to all products and is valued more highly than the certification label [12].

There are other certification benchmarks that have not been mentioned, but which are also important for environmental sustainability, such as integrated production (management of natural resources, favoring natural regulation, control of agrochemicals used, and safety times), organic production (determining the type of agrochemicals used, favoring biodiversity, preservation of natural resources) [43] or the Global GAP (benchmark for good agricultural practices) [44]. These models only focus on agricultural practices, but they are also applicable to viticulture.

## 6. Method

In June 2023, we conducted a literature review on environmental sustainability in wine cooperatives and their difficulties in responding to the new demands of markets and governments. For this analysis, we used two databases, ScienceDirect and Scopus, employing keywords and various combinations of them, i.e., sustainability, environment, cooperative wineries, cooperativism, sustainability benchmarks, and indicators. Figure 5 shows the research strategy. First, the word "sustainability" was included; then "cooperative" was included, then "environment", and finally different variables were inserted. Some restrictions were imposed: years of publication between 2015 and 2013; only research and review articles; and environmental, agricultural, and social areas. The search resulted in 2628 articles (1850 articles in ScienceDirect, 778 articles in Scopus), from which 126 articles were extracted for analysis. The rest were rejected because they were not associated with the wine sector or cooperativism or environmental sustainability, and because of duplication. Finally, 27 articles were included in this study. No software was used to support the analysis.

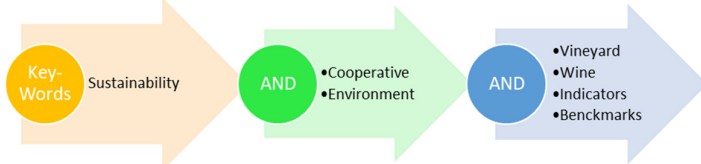

**Figure 5.** Search strategy use. Different keyword combinations. For example: sustainability AND cooperative AND indicators.

## 7. Discussion

Cooperatives have an important role in agriculture, but also in the communities in which they are established. The origins of cooperative agricultural organizations are associated with moments of crisis, when small producers join forces to sell the farm products they produce [13,45].

As we discussed at the beginning of this paper, companies and organizations must respond to the United Nations' challenge by creating benchmarks to meet the SDGs. Cooperatives have some of these goals as a priority, such as SDG 1, No Poverty. Cooperatives have been created to improve community conditions, such as SDG 2, No Hunger. In the case of agricultural cooperatives, the promotion of sustainable practices contributes not only to SDG 2, but also to SDGs 12 (Responsible Consumption) and 15 (Life and Land). According to the indicators analyzed above, cooperatives should be able to respond to SDGs 4 (Quality Education), 5 (Gender Equality), 8 (Good Jobs and Economic Growth), and 10 (Reduce Inequalities) through training, improving working conditions, and promoting gender and pay equality. The implementation of measures to mitigate climate change, such as water management and the use of renewable energy sources, should respond not only to SDGs 12 and 15, discussed above, but also to SDGs 6 (Clean Water and Sanitation), 7 (Renewable Energy), and 13 (Climate Action). Cooperatives are always well integrated into the community, and often provide support, so they always create synergies with government, social, political, business, and educational/research institutions. Since their mindset involves overcoming difficulties, administrations are very receptive to innovation. After this brief analysis, we can say that they easily respond to SDGs 9 (Innovation and Infrastructure) and 17 (Partnership for the Goals). Figure 6 below represents the SDGs that can be met by cooperatives if they implement the indicators discussed in the previous point.

In a quick analysis of the indicators listed in the benchmarks studied, it is possible to see that they respond to practically all of the SDGs in a more or less exhaustive way, as shown in Figure 7.

At the beginning of this review, some questions were raised, and with the information that has been compiled, we will try to answer them.

### 7.1. Q1: What Is Necessary to Achieve Sustainability?

First, we need to define sustainability, which, according to Ferrer [2], is the adaptation of human activities to guarantee the future of the next generations. In other words, it means securing a future where climate change has little impact, but also economic and social stability.

Analyzing the different benchmarks for sustainable certification, we were able to suggest a broad range of indicators that are common to the various models. These indicators address not only environmental issues, but also economic and social ones. For example, the Swedish market, Systembolaget [26], not only values environmental indicators but also gives great importance to social indicators, such as fair remuneration, non-discrimination, and precarious labor.

### 7.2. Q2: Is the Wine Cooperative Competitive in the Global Market?

Like any other company or organization, the cooperative is equally exposed to market challenges. The business model has proved resilient in times of crisis [12,13]. The objective of cooperatives is to sell the products of their members, remunerate them as much as possible, and reinvest the profits. However, this depends on the governance model and the members' commitment to the cooperative, for which they must maintain a high level of satisfaction. However, we have not fully answered the question because the challenges of market sustainability are what is needed. In the area of social sustainability, cooperatives respond comfortably, since this is the genesis of their creation, as well as their own economic sustainability and that of the community in which they are inserted. If the question is asked to each producer individually, it is not possible to answer because there is a lack of documentation. In the case of environmental sustainability, the producer is more attentive, although they are not sensitive to some indicators and do not measure them. There are

other factors that the producer monitors for legal reasons or economic interests; for example, to comply with the Integrated Production Mode or the Organic Production Mode [43].

Even if the cooperative is competitive at the moment, it must create tools to respond to sustainability criteria, because the market demands it and consumers are becoming increasingly aware of these issues.

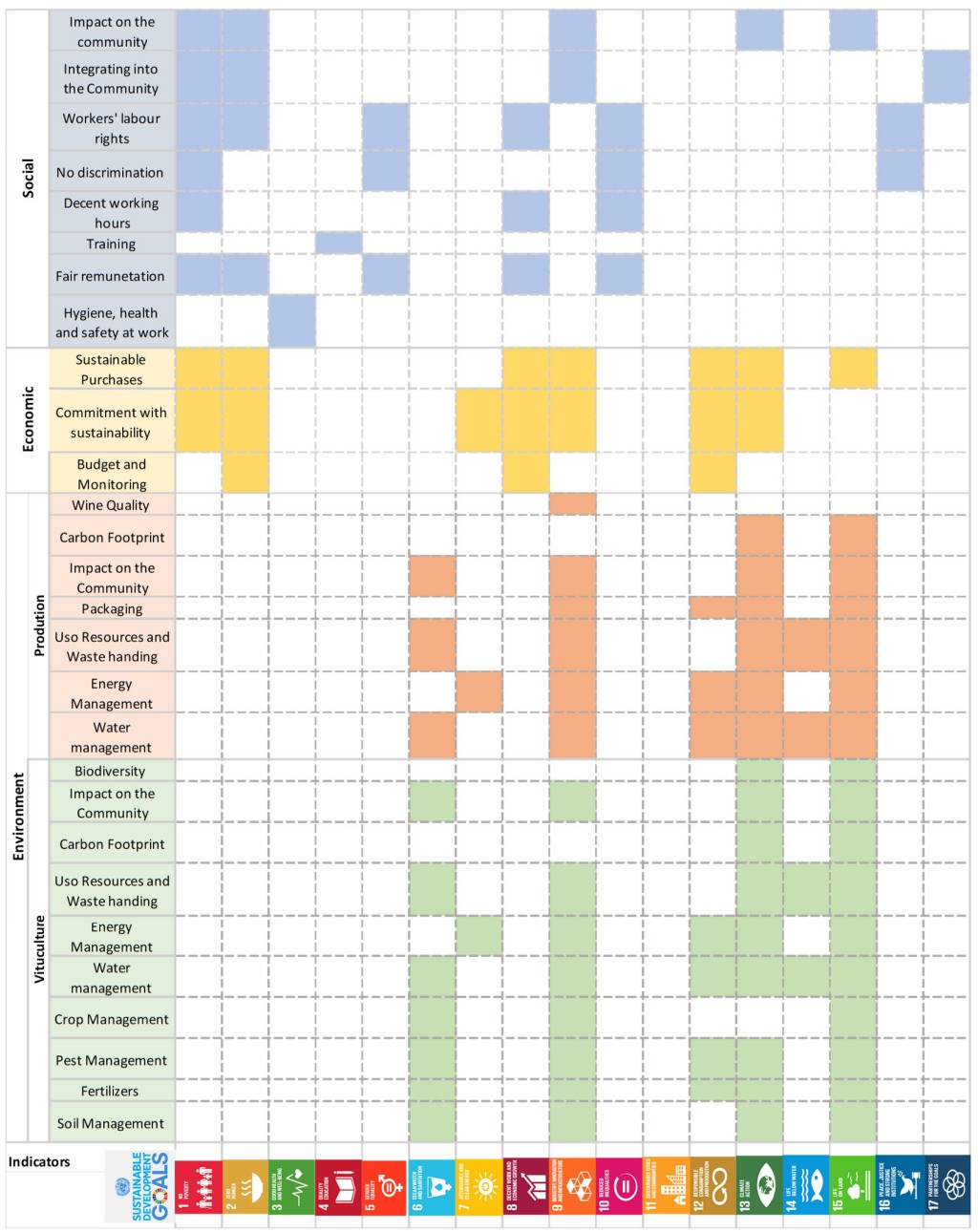

**Figure 6.** List of indicators identified in the benchmarks and their relationship with the SDGs (legend: green—indicators mentioned in the benchmarks for viticulture; orange—indicators mentioned in the benchmarks for wine production; yellow—economic sustainability indicators mentioned in the benchmarks; blue—social sustainability indicators mentioned in the benchmarks).

### 7.3. Q3: How Can We Respond to the Challenges of Environmental Sustainability While Maintaining Wine Quality Standards and Economic Profitability?

Although the literature explores environmental sustainability, it was not possible to find a relationship with quality and economic profitability in the cooperative model. The literature shows studies on the economic sustainability of cooperatives and their model,

with its advantages and difficulties. However, when we tried to analyze whether the impact of environmental measures has a positive or negative economic impact, these data were not shown. We are unable to conclude whether some environmental measures have not been implemented for financial reasons, or if their implementation could present a cost reduction that would be attractive to the producer. To answer this question, there needs to be more research into the effect of measures to reduce the environmental impact on economic sustainability and also their economic viability, such as the effect on wine quality. Soregaroli [7], in a consumer survey, found that they valued the economic factor more than the carbon footprint. However, if the customer has the perception that wine with a low carbon footprint has higher quality, they will choose it [7]. Ferrer [2] analyzed the business model of 411 wineries in Spain and devised two types of business model: highly sustainable and low sustainability [2]. The major difference in the model was only related to the fact that the highly sustainable business model had a well-defined structure, favored the sale of bottled wine, and had knowledge of the entire process [2]. The model with a low level of sustainability sold mostly in bulk and did not know where the wine was going [2].

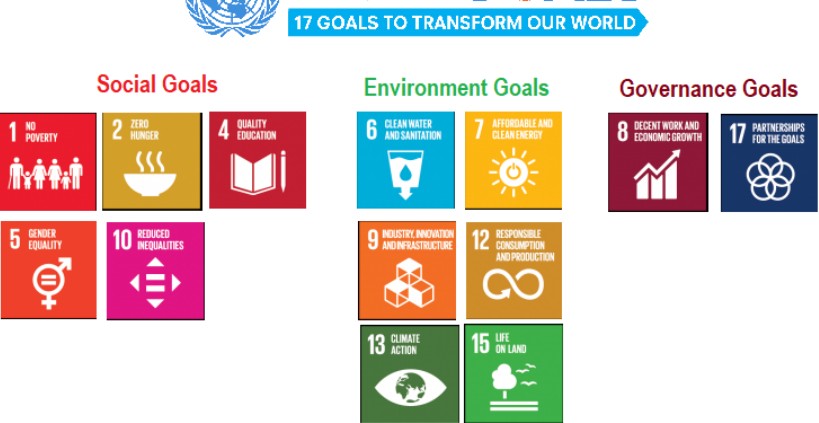

**Figure 7.** Sustainable development objectives that can be answered by wine cooperatives.

*7.4. Q4: What Are the Economic and Social Impacts of Reducing the Carbon Footprint of the Winery and Its Members?*

In the same way that the literature did not answer the previous question, we did not find any answers in the literature to this question either. Cooperatives, especially wine cooperatives, are made up of small producers, most of whom are older and have low literacy levels, which makes it difficult not only to communicate but also to obtain answers, as Figueiredo [13] mentions in his study of wine cooperatives in the Dão wine region. However, there are no works in the literature that answer this question in the case of other types of organizations.

It is possible to have a consistent and comprehensive group of sustainability indicators, already implemented and with a track record in the wine sector, but there is a lack of studies on the impact of these indicators on communities, organizations, and consumers. It is important for small producers to realize that they have a fundamental role to play on the road to sustainability, but they need to know what the economic advantage is. Their priority is to satisfy their needs, and selling their products to the cooperative will fulfil them.

## 8. Conclusions and Future Directions

To answer the questions raised, it is necessary to develop a methodology that allows wine cooperatives to calculate their level of sustainability in a credible way, as well as that of their members. This methodology should cover the most relevant indicators: water management, soil management, vine management (including crop practices, nutrition, and pest control), energy management, carbon footprint, and human resources (workers' rights,

hygiene, health, safety at work). It should also respond via the cooperative organization to indicators on local biodiversity and the impact of activities on the community (not only environmentally, but also socially).

Through the analysis of the dynamics of cooperative wineries, we can transform weaknesses into added value, such as by giving members an active role in sustainability, creating integration tools to mitigate economic differences such as financial capacity or the area of land parcels. Providing the organization with tools with which they can integrate their members will enable them to respond to the current environmental, economic, and social challenges not only imposed by the wine markets, but also by the current socio-economic situation. In other words, this will create activities by which the environmental, social, and economic aspects of winegrowing members and wine production can be improved.

**Author Contributions:** Conceptualization, C.A.T. and A.M.; methodology, C.A.T.; software, A.M.; validation, C.A.T. and A.M.; formal analysis, A.M.; investigation, A.M.; resources, A.M.; data curation, C.A.T.; writing—original draft preparation, A.M.; writing—review and editing, C.A.T.; visualization, A.M.; supervision, C.A.T.; project administration, C.A.T.; funding acquisition, C.A.T. All authors have read and agreed to the published version of the manuscript.

**Funding:** This work was supported by National Funds from the FCT—Portuguese Foundation for Science and Technology, under the project UIDB/04033/2020.

**Data Availability Statement:** The data used to support the findings of this study are available from the corresponding author upon request.

**Acknowledgments:** We thank Adega Cooperativa de Favaios, CRL, for their support in the development of the project.

**Conflicts of Interest:** The authors declare no conflict of interest.

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
