# Peer review of "Vine and Wine Sustainability in a Cooperative Ecosystem—A Review"

_agronomy, doi:10.3390/agronomy13102644_

Round 1
Reviewer 1 Report
The paper is well written and the review done is comprehensive of the major and relevant standards in the field. I have truly appreciated the degree of systemic analysis adopted.
The main research question in this paper is to provide a comprehensive review of the main protocols and certifications in the field of sustainability for the wine and vine industry. The topic is not futuristic, but I do think that the study grasps a relevant need for practitioners and scholars in other disciplines outside of agronomy to understand the field of sustainability for wine and vine. In addition, the paper is grounded in a cooperative ecosystem shedding light on the relevant features of such a kind of meta-organizations. The focus on the cooperative ecosystem is the value added to this work, as there is a paucity of studies on the nexus between food science and cooperativistic models. The paper is actually missing a section dedicated to the methodology, as it seems more like a qualitative review. More should be added on how data has been coded in revising the type of certifications that exist and how this has been triangulated with the data on the field of cooperative sector. A greater emphasis on the interplay between cooperativism and certifications could be stresses, maybe in relation to several crucial issues such as: undeclared workers, supply chain issues, health and safety The references are appropriateI have no additional comments on the tables and figures.
My major comment is about the need of stress more on the social implications of the impacts and also on the need of making a focus on SMEs difficulties and family business aspects in vine and wine sector.
The paper is well written and clear
Author Response
We agree with the comments and that is why we added figure 6 and improved figure 7.
Reviewer 2 Report
The paper makes an extensive analysis of the dynamics of cooperative wineries, where an attempt is made to transform weaknesses into added value, such as providing members with an active role in sustainable development, creating tools to mitigate economic disparities, such as financial opportunities. The current environmental, economic and social challenges are described extensively. The work is very interesting at a high level. I recommend publishing the work.
The paper makes an extensive analysis of the dynamics of cooperative wineries, where an attempt is made to transform weaknesses into added value, such as providing members with an active role in sustainable development, creating tools to mitigate economic disparities, such as financial opportunities. The current environmental, economic and social challenges are described extensively. The work is very interesting at a high level. I recommend publishing the work.
Author Response
Thank you very much for your comments. They are a source of positive energy and confidence for the challenges that lie ahead.
Reviewer 3 Report
Dear authors,
I am pleased of having the opportunity to review such an interesting research paper, which provides a comprehensive review of the challenges and opportunities facing wine cooperatives in the context of sustainability, delving into the historical importance of cooperatives, their principles, and the current challenges they face in the wake of climate change. The arguments are supported with relevant literature, making it a valuable resource for those interested in the interaction of sustainability, wine production and cooperatives.
In this regard, I have some comments that may help to improve the quality of the paper:
· The title contains the phrase "A Accurate Review", which is grammatically incorrect. It should be corrected to "An Accurate Review".
· The abstract provides a general overview but lacks specific details about the methodology, key findings, or implications of the research. An ideal abstract should give a concise summary of the entire paper, including its main conclusions.
· While the introduction mentions the UN's Sustainable Development Goals, it would be beneficial to provide more context on how these goals specifically relate to the wine industry and cooperatives.
· The transition between sections could be smoother. For instance, the jump from the historical context of wine production to the specifics of cooperatives feels abrupt. A bridging paragraph or statement could help in maintaining a logical flow.
· Figure 1: A more detailed caption or description would help readers understand the relevance of the Sustainable Development Goals to the paper's topic.
· Table 1 & Table 2: The tables provide literature reviews but lack a clear explanation of the criteria for inclusion or the significance of the works cited. A brief overview or rationale for the selection of these works would enhance the paper's depth.
· The challenges faced by cooperatives are discussed, but potential solutions or recommendations are not provided. Addressing these challenges and offering insights or suggestions would add value to the paper.
· While the challenges are highlighted, the paper could delve deeper into the specific sustainability practices that cooperatives are currently implementing and their effectiveness.
· There are several grammatical errors throughout the paper. A thorough proofreading is essential to enhance the paper's readability and professionalism.
· The paper seems to lack a strong conclusion that summarizes the main findings and offers future directions or implications for the wine cooperative industry in the context of sustainability.
I am waiting for the modified version, I am sure that addressing these aspects will improve the quality of the work significantly.
Greetings,
no comment
Author Response
Thank you very much for your comments. We change the title and we seek to respond to your challenges in chapter 6. Method and chapter 7. Discussion. Therefore we added figures 6 and 8 and improved the figure 7.